# Validating a Traffic Conflict Prediction Technique for Motorways Using a Simulation Approach [note 1]

**DOI:** 10.3390/s22020566

**Published:** 2022-01-12

**Authors:** Nicolette Formosa, Mohammed Quddus, Alkis Papadoulis, Andrew Timmis

**Affiliations:** 1School of Architecture, Building and Civil Engineering, Loughborough University, Loughborough LE11 3TU, UK; n.formosa@lboro.ac.uk (N.F.); a.j.timmis@lboro.ac.uk (A.T.); 2Department of Civil and Environmental Engineering, Imperial College London, London SW7 2AZ, UK; 3Immense Simulations Limited, London EC1A 2BN, UK; alkis.papadoulis@immense.ai

**Keywords:** validation, integrated simulation framework, traffic conflicts, road safety

## Abstract

With the ever-increasing advancements in the technology of driver assistant systems, there is a need for a comprehensive way to identify traffic conflicts to avoid collisions. Although significant research efforts have been devoted to traffic conflict techniques applied for junctions, there is dearth of research on these methods for motorways. This paper presents the validation of a traffic conflict prediction algorithm applied to a motorway scenario in a simulated environment. An automatic video analysis system was developed to identify lane change and rear-end conflicts as ground truth. Using these conflicts, the prediction ability of the traffic conflict technique was validated in an integrated simulation framework. This framework consisted of a sub-microscopic simulator, which provided an appropriate testbed to accurately simulate the components of an intelligent vehicle, and a microscopic traffic simulator able to generate the surrounding traffic. Results from this framework show that for a 10% false alarm rate, approximately 80% and 73% of rear-end and lane change conflicts were accurately predicted, respectively. Despite the fact that the algorithm was not trained using the virtual data, the sensitivity was high. This highlights the transferability of the algorithm to similar road networks, providing a benchmark for the identification of traffic conflict and a relevant step for developing safety management strategies for autonomous vehicles.

## 1. Introduction

With the rapid growth of artificial intelligence techniques such as machine learning and image recognition, the vehicle industry is currently undergoing dramatic changes. A vehicle is no longer considered to be a simple mechanical structure, but an entity which includes multiple advanced driver assistance systems (ADAS) aiming to improve both vehicle and driver safety. With these additional functions, vehicles are progressively becoming more intelligent, with the latest advancement being Autonomous Vehicles (AVs) and Connected and Autonomous Vehicles (CAVs). These emerging technologies in the automotive industry have introduced safety-related challenges and, consequently, research has attempted to address these challenges by developing proactive and predictive safety management systems for these vehicles.

Traditionally, safety analysis has formed functional relationships between occurrences of safety incidents (e.g., accidents) and various kinematics or infrastructure related factors using historical data [1]. However, these approaches have been developed by static and aggregated data, and the effects of sudden changes in traffic conditions that lead to collision risks are sometimes not captured [2]. These limitations can be overcome by making use of real-time traffic data [3]. This approach is considered to be proactive rather than reactive [4]. The underlying theory behind the approach of using non-accident data is based on that in accelerating vehicle-based proactive safety systems, traffic conflicts can be used a measure of accident nearness, and a safety critical event has the potential to progress to a traffic collision. In fact, Tarko, [5] establishes that traffic conflicts are probabilistically and statistically connected with traffic accidents. Typically, a traffic conflict includes including at least two moving vehicles and is generally identified through the response of the evasive action that takes place so as to avoid a potential collision, and by a safety surrogate measure (SSM) that shows a spatio-temporal risk (two parties are in close proximity in either space or time) [5,6,7]. Nevertheless, reliable traffic conflict identification is not an easy task.

Traffic conflict prediction algorithms have been developed to assess and quantify the threat level surrounding an ego-vehicle. The technical reliability of adopting such systems for conflict prediction relies on the functionality of the multiple sensors that a vehicle is equipped with. These systems necessitate stringent safety tests to ensure robustness. Testing and validation is key to determining the prediction accuracy and impact on safety that a traffic conflict prediction algorithm can achieve in a complex and organically changing real-world environment, thereby facilitating customer uptake. Commercially available sub-microscopic simulators such as PreScan, CarMaker, and CARLA provide platforms offering a variety of intelligent traffic modeling to investigate and validate various technologies and applications such as testing the identification of traffic conflicts. This can be done by including a traffic conflict prediction model using deep learning within this simulator, so that complex and non-linear patterns are identified to classify and predict traffic conflicts [8]. To boost the strength of sub-microscopic simulators, they can also be integrated with other available simulation tools such as PTV VISSIM (a microscopic simulator). However, the complexity of modeling all possible combinations of traffic situations and environmental conditions make this approach challenging. Hence, a number of ‘benchmark’ scenarios have to be developed allowing the investigation of driving behaviors and vehicle functionalities under different traffic conditions and scenarios.

Therefore, the purpose of this paper is to validate a traffic conflict prediction algorithm implemented within an integrated sub-microscopic simulation framework. One main challenge arising from validating the prediction algorithm is that ground truth, i.e., traffic conflicts, is not readily available. This study makes use of a system that adopts a set of defined conservative criteria derived from existing literature to identify all possible traffic conflicts. To enhance the validation of ground truth, the results from this system are also validated. The ground truth provides a means for virtual verification and evaluation of the traffic conflict predictions from the algorithm in critical scenarios using an integrated simulation framework. To ensure realistic values and accurate representation of traffic characteristics, data was collected via an instrumented vehicle and inductive loop detectors from a section of the M1 motorway in the UK. The scenarios were developed based on several variables including the ego-vehicle and the behavior of the surrounding vehicles, the sensors used, and the driving style of road users. These variables need to be modeled in a sub-microscopic simulation environment while a microscopic traffic simulator can be used in conjunction to generate traffic dynamics.

This research is a valuable addition to existing literature, since it develops a process to acquire traffic conflict data to test the effectiveness of the prediction algorithm by developing a number of traffic conflict scenarios in the simulation framework. The integrated simulation framework is a new approach that provides a platform to the user to model, test, validate, and further enhance the customized models for future users. The results emerging from the framework showcase the transferability of the algorithm and provide recommendations to vehicle manufacturers to improve their collision-risk model, allowing intelligent vehicles to operate safely and reliably, and further testing and developing vehicle safety technologies and automated driving systems.

## 2. Literature Review

Timely detection of traffic conflict plays an important role in the development of intelligent vehicles in order to proactively mitigate the risk of collision. Multiple traffic conflict identification techniques have been recommended and applied in the literature, but these studies provide mixed conclusions about their transferability to motorway scenarios. As a result, this review focuses on two key parts. Firstly, it examines the characteristics used to identify ground-truth traffic conflicts occurring in a motorway scenario. It must be noted that only lane change and rear-end traffic conflicts are considered in this study, as these are the two most prevalent type of traffic conflicts in a motorway scenario [9]. Secondly, it presents a review of existing integrated simulation frameworks to test and validate a traffic conflict predictor algorithm. These frameworks assist to improve safety in ADASs and highlight their utility in CAVs and AVs.

### 2.1. Traffic Conflict Identification to Generate Ground Truth Data in a Motorway Scenario

The identification of traffic conflicts is a challenging task. They are characterized by a set of indicators arising from their definition: a spatiotemporal proximity indicator and an evasive action response [7]. These indicators can facilitate the identification process and research has attempted to define thresholds that clearly delineate the presence of a traffic conflict. However, these thresholds have yet to be solidified.

Within the evasive action, there are three possible corrective maneuvers an ego-driver can take in response to a traffic conflict: control speed by (1) accelerating or (2) braking [10], and/or (3) steering to control lateral position [11]. By analyzing these evasive maneuvers, information on how they are performed can be derived (e.g., controlled braking, lane changing), as well as how they are influenced by the dynamic surroundings or road-surface conditions in which they take place. While some traffic conflicts can be identified using kinematic triggers, this can lead to a selection bias, since traffic conflicts can also result in no reaction from the driver and therefore no significant kinematic changes [12]. This highlights that the identification of traffic conflicts should fundamentally not be based solely on evasive maneuvers, but also on the temporal and/or spatial proximity of the vehicles [7]. Examples of temporal proximity measures include time headway and Time-to-Collision (TTC) whilst spatial proximity measures include examples such as Proportion of Stopping Distance (PSD). The key advantage of spatial and/or temporal proximity measures is their ability to capture the severity of a traffic conflict interaction in an objective and quantifiable way [13]. However, this assumption may not be applicable in all driving environments, for example, in congested traffic settings where road users are predisposed to regular vehicular interactions and are frequently near each other.

Therefore, by adopting both evasive maneuvers and spatial and/or temporal proximity measures, traffic conflict identification can be improved via information about the ego-vehicle and surrounding traffic. Consequently, in this way, traffic conflicts can be identified and assessed accurately and reliably. Nevertheless, detection of the traffic conflict indicators from vehicles and trajectories via vision-based monitoring of road networks is inherently difficult [14]. This approach is typically followed in existing studies for intersections using static video data [15,16]. However, when using static video data, inherent challenges arising from the development of the latest technology advancements, such as autonomous vehicles, are not covered. Video data collected from an in-vehicle perspective could add a new real-time dimension to traffic conflict identification analysis by examining a continuously changing environment. Therefore, such a dynamic video analysis system needs to be developed to provide valid and reliable traffic conflict identification for a continuously changing visual and driving environment.

### 2.2. Testing and Validation of Functions in Intelligent Vehicles

The development of systems for intelligent driving functions is a complex process. The technical reliability of intelligent vehicles relies on the functionality of the multiple systems the vehicle is equipped with. The system’s performance changes with road infrastructure and dynamic surroundings. Hence, testing and validation of systems are of great importance. It is necessary that comprehensive testing, either on real-world test tracks and/or in virtual environments, is performed. However, most testing methods are still carried out virtually rather than on public roads [17]. Virtual testing includes making use of traffic simulation software programs where each part of the system (e.g., road network, vehicle systems and driving behavior) is modeled. It is necessary to design, explore and evaluate novel emerging transportation systems. These are the only methods that can address unprecedented challenges that will arise from their implementation safely and at a low-cost. There are four categories of traffic simulation: macroscopic, mesoscopic, microscopic, and sub-microscopic. Each category depends on the detail of simulation required and the scale of the experiment that needs to be conducted.

The lowest level of detail is found in macroscopic and mesoscopic models. These models are mainly used to explore travel demand between origin and destination. Following these models, changes in traffic patterns are explored in microscopic traffic simulation models such that they are able to reproduce the behavior of individual vehicles. In fact, traffic microsimulation makes use of mathematical equations to represent the movement of simulation vehicles. Examples of microscopic simulation software tools include PTV VISSIM, AIMSUN, or SUMO (open-source). One of the main drawbacks of traffic microsimulation is the fact that it cannot accurately simulate specific mechanical or technological parts of a system. As a result, it cannot assist in addressing specific challenges associated with sensor recognition faults, such as the failure to identify lane markings or a roadworks area in a motorway.

Sub-microscopic simulation can accurately simulate all the components of an intelligent vehicle such as sensors (e.g., camera and radar), mechanical parts, and internal components (e.g., engine, steering wheel). This category has also some disadvantages; namely, it is not able to manage large-scale experiments with regards to network size and cannot simulate large amounts of surrounding traffic. It works by using the graphical user interface (GUI) of a sub-microscopic simulator to model and outline scenarios. Control algorithms for each component (e.g., vehicle or infrastructure) are developed using C++, MATLAB, or Python. Typical sub-microscopic software programs used are: (1) PreScan, (2) CarMaker, and (3) CARLA. A summary of how these sub-microscopic simulators have been adopted in the literature for virtual testing for safety purposes is presented here:PreScan:

Real-world tests were translated to virtual driving in a simulation environment by Van der Made et al. [18] to evaluate different scenarios. Validation of probabilistic ADAS sensor models with real-world data developed by Schubert et al. [19] was also carried out in PreScan. On the other hand, Seo et al. [20] adopted a virtual sensor model together with control algorithms to compare autonomous emergency braking with a pre-safe seat belt system;

CarMaker:

Riegger et al. [21] developed and validated a system for safe crossing. The system took control over AVs to optimize their trajectories. Überbacher et al. [22] evaluated lane change warning and side collision warning by applying a vehicle-in-the-loop function. Pfeffer and Haselhoff [23] tested the performance of a camera-based driving assistance system by adding artificial pedestrian models into the video stream;

CARLA:

Osiński et al. [24] used interactive traffic scenarios to train agents to develop policies for a variety of autonomous driving systems. These scenarios were also used for experimental evaluation. Dosovitskiy et al. [25] employed CARLA to compare the performance of three methods for autonomous driving. These methods include the classical model, a model trained via imitation learning, and a model trained via reinforcement learning.

A variety of these simulators could potentially be used in this study; however, the candidate software must achieve several essential requirements. These requirements review the tools’ capability to simulate vehicle equipment, perform realistic maneuvers, and accurately simulate the vehicle’s surrounding environment and infrastructure. The inventory of all the available traffic sub-microscopic simulators along with the criteria are presented in Table 1.

As a result, from Table 1, PreScan was the simulation tool adopted for this study. This is because of its ability to integrate the essential multiple components of a traffic model, including vehicles (i.e., intelligent mobility functionalities), human behavior, and traffic network via user interfaces. PreScan can also simulate multiple sensor technologies such as cameras, LiDARs, radars, and GPS. It operates by employing Simulink, enabling the representation of sophisticated system/scenario control algorithms, the development of new algorithms, and modifications of existing ones. The main strength of this platform is its ability to offer a variety of intelligent traffic modeling in order to design and investigate various technologies and applications.

In conclusion, testing and validation of any ADAS is crucial. Existing studies have investigated the safety effectiveness of different ADAS functions in various simulators [19,20,21,22,23,24,25]. However, ground truth is not always known a priori. In this study, to extract a potential traffic conflict, criteria derived from existing literature are used to include all traffic conflicts. Therefore, using these criteria, an extraction algorithm is adopted to identify all time points where the thresholds were exceeded. Nevertheless, the time points identified by the system need to be validated. By comparing the extracted traffic conflicts ground truth with traffic conflict predictions from the implemented algorithm, the technical reliability and performance of the algorithm can be evaluated. Such processes can be performed in an integrated simulation framework. In this research, the framework considered consists of a submicroscopic simulator (PreScan), which provides the appropriate testbed and can accurately simulate the components of an intelligent vehicle, in conjugation with a microscopic simulator (PTV VISSIM) to simulate the surrounding traffic dynamics. This framework is considered due to its wide acceptability within the research community and its capability to test ADAS. This work is significant as it provides benchmarking for the prediction of traffic conflicts for virtual testing procedures and is a relevant step for developing safety management strategies for AVs.

## 3. Methods

The methodology of this paper is divided into two sections. The first section includes traffic conflict identification to generate ground truth data and the second section includes the validation of the traffic conflict prediction algorithm. Identifying traffic conflicts ground truth is not easy as the data is not readily available. As a result, to identify rear-end and lane changing conflicts, a method must be developed in a needed simulated environment. A system making use of vehicle detections, lane-geometry identification, and defined criteria was developed virtually to identify traffic conflicts. For each conflict scenario, safety metric equations were developed to evaluate the traffic conflict identification.

An empirical analysis was carried out on this data to identify traffic conflict defining patterns as well as the nature of performed evasive maneuvers when a traffic conflict is identified. Based on this empirical analysis, derived scenarios were then transferred to an integrated simulation framework to investigate safety performance. The simulation framework consists of a submicroscopic and a microscopic traffic simulation tool. The submicroscopic simulator served as an appropriate testbed to validate and test the traffic conflict predictions from the developed algorithm implemented within the ego-vehicle. The microsimulation tool was utilized to simulate dynamic traffic surrounding the ego-vehicle.

### 3.1. Generation of Traffic Conflicts Ground Truth from Simulated Data

The video analysis system was developed to identify traffic conflicts through the response to an evasive action and the temporal and/or spatial proximity [7]. A set of defined conservative criteria derived from existing literature were used to include all traffic conflicts. Therefore, using these criteria, an automatic extraction algorithm was developed to identify all the vehicles and time points where the thresholds were exceeded. The influential factors used to identify both lane change and rear-end conflicts were based on time, distance, speed, braking, acceleration and deceleration. The readers are referred to Formosa et al. [26] for a more thorough description of the traffic conflict extraction from the data. The flowchart for the lane change and rear-end conflict extraction algorithm is shown in Figure 1.

These traffic conflicts were further validated by evaluating developed equations for both traffic conflict scenarios. These equations are based on the assumption that in a conflict process, the preceding vehicle (or the lane changing vehicle) maintains its speed while the ego-vehicle applies the brakes once the driver reacts. This assumption is reasonable, since the drivers in the front are possibly not aware of the hazard conditions behind them [27]. Considering the lane change conflict presented in Figure 2a, if the two vehicles maintain their velocity, the conflict will turn into a sideswipe conflict at time t_X_ with angle θ, and the following conditions need to be satisfied:(1)$$s_{{\mathrm{EV}}_{\mathrm{LC}-\mathrm{SS}}}\leq V_{\mathrm{EV}}t_{X}+s_{\mathrm{LC}}+s_{\mathrm{LCV}}-\left(\frac{l_{\mathrm{EV}}+l_{\mathrm{LCV}}}{2}\right)$$
(2)$$V_{\mathrm{EV}}\left(t_{X}+t_{\mathrm{LC}}\right)-\frac{a_{\mathrm{EV}}}{2}{\left(t_{X}+\frac{l_{\mathrm{LCV}}}{V_{\mathrm{LCV}}}-\tau_{\mathrm{LC}}\right)}^{2}\leq \;V_{\mathrm{EV}}t_{X}+\frac{w_{\mathrm{LCV}}}{2\sin\theta }+\frac{w_{\mathrm{EV}}}{2\tan\theta }+\frac{l_{\mathrm{LCV}}(1+\cos\theta )}{2}-\left(\frac{l_{\mathrm{EV}}+l_{\mathrm{LCV}}}{2}\right)$$

In this case, the required braking rate (RBR) is given as:(3)$$a_{\mathrm{LC}-\mathrm{SS}}\geq \;\frac{\frac{2V_{\mathrm{EV}}l_{\mathrm{LCV}}}{V_{\mathrm{LCV}}}+l_{\mathrm{EV}}-l_{\mathrm{LCV}}\cos\theta -\frac{w_{\mathrm{LCV}}}{\sin\theta }-\frac{w_{\mathrm{EV}}}{\tan\theta }}{{\left(t_{X}+t_{\mathrm{LC}}-\tau \right)}^{2}}$$
where τ is the reaction time taken as 1.3 s [28]; t_LC_ is the time for the lane change vehicle to enter the ego-vehicle’s lane, which can be calculated by t_LC_ = l_LCV_⁄V_LCV_ [27] where l_LCV_ and V_LCV_ is the length and velocity of the lane change vehicle; l and w are the length and width of the vehicle, respectively; and t_X_ is the modified TTC. This is estimated by $\frac{-\Delta V\;\pm \surd \left({\Delta V}^{2}+2\;\Delta a\;\Delta X\right)}{\Delta a}$, where ΔX is the clearance between two vehicles and ΔV and Δa denote the rate of change in velocity and acceleration between the ego-vehicle and the lane change vehicle, respectively. However, when the conflict angle varies between [0°, 5°], the sideswipe conflict is also considered as a rear-end conflict resulting in the following equation:(4)$$s_{{\mathrm{EV}}_{\mathrm{LC}-\mathrm{RE}}}\leq V_{\mathrm{EV}}t_{X}+s_{\mathrm{LC}}^{\prime }+s_{\mathrm{LCV}}^{\prime }-\left(\frac{l_{\mathrm{EV}}+l_{\mathrm{LCV}}}{2}\right)$$
(5)$$V_{\mathrm{EV}}\left(t_{X}+t_{\mathrm{LC}}\right)-\frac{a_{\mathrm{EV}}}{2}{\left(t_{X}+\frac{l_{\mathrm{LCV}}}{V_{\mathrm{LCV}}}-\tau_{\mathrm{LC}}\right)}^{2}\leq \;V_{\mathrm{EV}}t_{X}+\frac{w_{\mathrm{LCV}}}{2\sin\theta }+\frac{w_{\mathrm{EV}}}{2\tan\theta }+\frac{l_{\mathrm{LCV}}(1+\cos\theta )}{2}+V_{\mathrm{LCV}}\left(\tau +\frac{V_{\mathrm{EV}}-V_{\mathrm{LCV}}}{a_{\mathrm{EV}}}-t_{\mathrm{LC}}\right)-\left(\frac{l_{\mathrm{EV}}+l_{\mathrm{LCV}}}{2}\right)$$

The RBR for this condition is:(6)$$a_{\mathrm{LC}-\mathrm{RE}}=\frac{{\left(V_{\mathrm{EV}}-V_{\mathrm{LCV}}\right)}^{2}}{2\left(V_{\mathrm{EV}}\left(t_{X}-\tau \right)+V_{\mathrm{LCV}}\tau +\frac{w_{\mathrm{LCV}}}{2\sin\theta }+\frac{w_{\mathrm{EV}}}{2\tan\theta }+\frac{l_{\mathrm{LCV}}\cos\theta -l_{\mathrm{EV}}}{2}-l_{\mathrm{LCV}}\right)}$$

In the case of the rear-end conflict presented in Figure 2b, if the ego-vehicle is not able to decelerate to at least the speed of the preceding vehicle with a clearance of 0 m, a conflict will take place.
(7)$$s_{{\mathrm{EV}}_{\mathrm{RE}}}\leq s_{\mathrm{EV}-\mathrm{PV}}+s_{\mathrm{PV}}-\left(\frac{l_{\mathrm{EV}}+l_{\mathrm{PV}}}{2}\right)$$
(8)$$V_{\mathrm{EV}}\tau_{\mathrm{RE}}+\frac{V_{\mathrm{EV}}^{2}-V_{\mathrm{PV}}^{2}}{2a_{\mathrm{EV}}}\leq V_{\mathrm{EV}}t_{X}+\left(\frac{l_{\mathrm{EV}}+l_{\mathrm{PV}}}{2}\right)-V_{\mathrm{PV}}t_{X}+\frac{V_{\mathrm{PV}}\left(V_{\mathrm{EV}}-V_{\mathrm{PV}}\right)}{a_{\mathrm{EV}}}-\left(\frac{l_{\mathrm{EV}}+l_{\mathrm{PV}}}{2}\right)$$
with the RBR of:(9)$$a_{\mathrm{RE}}\geq \;\frac{V_{\mathrm{EV}}-V_{\mathrm{PV}}}{2{\left(t_{X}-\tau \right)}^{2}}$$
where τ is the reaction time taken as 0.92 s [29].

### 3.2. Traffic Conflict Algorithm Validation

Validating the results from the developed algorithm is key to proving its effectiveness. This process can be performed in a traffic simulation environment, providing a platform where the safety performance of the prediction algorithm can be evaluated and validated. This approach improves the cost-effectiveness of the development stage and was achieved by allowing the maximum number of scenarios to be tested against different factors. An advantage such as this would not be feasible in real-world tests given the inherent complexities in creating such scenarios [17].

#### 3.2.1. Ontology of Simulation Experiments

In the current literature, the terminology used to describe the virtual evaluation and validation of ADAS development is inconsistent. An ontology presented by Geyer et al. [30] is adapted within this study to evaluate and validate the developed traffic conflict prediction algorithm (see Figure 3).

Figure 3 presents a hierarchical structure to show the different levels of the simulation study. These levels include:Experiment:

The top level of the hierarchical structure is an experiment in which a systematic procedure is conducted in order to evaluate and validate the developed traffic conflict prediction algorithm. This experimental level is described by the requirements needed for testing and validation, the network model, vehicles, study duration, and computational constraints;

Scenery:

This level describes all the static components of the network environment, i.e., the layout and geometry of the network to be simulated. This includes the addition of static components within the network, for example, road markings, traffic lights, and road signs;

Scenario:

A scenario introduces the dynamic traffic objects, i.e., road users. A set of scenarios were developed based on the objectives of the study using the scenery. Therefore, the scenario describes the interactions between the ego-vehicle and the surrounding traffic, their trajectories, position, and velocity to carry out a maneuver.

1.Ego-vehicle:

The sub-microscopic simulation requires a vehicle to be used as the main subject of investigation. The developed traffic conflict prediction algorithm is applied to this vehicle in order to be tested and validated. In comparison to the surrounding traffic, this vehicle is equipped with multiple sensors and has detailed models;

2.Criticality:

This defines a condition that must be examined by research. It is assumed that these conditions impact negatively the safety performance of the developed system to be tested. This may include adverse weather conditions, which may influence the sensor perception, and vision obstructions that lead to challenges for object detection.

#### 3.2.2. Simulation Framework for Validating a Traffic Conflict Prediction Algorithm

Traffic simulation paves the way for intelligent technologies by allowing virtual testing and validation of emerging ADAS in a realistic traffic environment. This research implemented traffic simulations to test and validate the developed traffic conflict prediction algorithm. Developing a simulation environment that can simulate intelligent mobility and its functions is crucial; therefore, PreScan was used. However, to boost the strength of the PreScan software, it was advantageous to integrate it with a traffic microsimulation software such as PTV VISSIM to create surrounding traffic environments. When both are applied simultaneously, a comprehensive integrated platform is built that is more powerful than the individual components. This allows for the investigation of driving behaviors and vehicle functionalities under different traffic conditions and scenarios.

The framework developed in this research consists of a sub-microscopic traffic simulation software (PreScan) that sends ego-vehicle information, while the microscopic traffic simulation software (PTV VISSIM) sends the information about the locations and kinematic characteristics of surrounding traffic. These two simulators exchange data through the Transmission Control Protocol/Internet Protocol (TCP/IP) during every simulation step (20 Hz). The architecture of the integrated simulation platform is summarized in Figure 4.

#### 3.2.3. Ontology of Simulation Experiments

A traffic conflict scenario depends on a number of factors, such as the ego-vehicle and the behavior of the surrounding vehicles, and the sensors employed and the driving style of road users. These factors need to be modeled in a sub-microscopic simulation environment in the network. Based on these factors, benchmark scenarios were developed. These scenarios are able to be completely quantifiable, controllable, and reproducible. Therefore, the prediction of lane change conflicts, rear-end conflicts, and a mixture of both conflicts from the developed prediction algorithm can be evaluated and validated. Based on the criteria discussed for lane change conflicts and rear-end conflicts (see Section 3.1), several scenarios were formulated.

Scenario development follows a four staged process, as represented in Figure 5. Firstly, the network for the study area is built using an aerial photograph. This includes building the derived scenario in the PreScan GUI, where Open Street Map and Google Earth images are used as underlays to create the road network. Besides the static road information, other parameters, such as the ego-vehicle model, do not vary between the scenarios. The environmental sensors of the ego-vehicle system are then modeled (e.g., radar, laser, camera, infra-red, GPS) within the submicroscopic simulator. The sensor design and benchmarking are made easier by modification of the sensor type and sensor characteristics from the PreScan GUI.

Subsequently, PTV VISSIM is used to represent the network section to generate the surrounding traffic flow based on the real data. The network development involves defining model parameters, vehicle composition, the number of lanes, the required input traffic data, and driving behavior characteristics. Nevertheless, for the simulation model to replicate real-world road traffic conditions, calibration of the road network and simulation parameters are required. This is carried out to minimize the deviation between observed field data and simulated data to prevent any misleading conclusions.

Typical parameters requiring calibration include simulation period, resolution, random seed number, and simulation runs [31]. These parameters are initiated with the default parameters set by VISSIM and real-world traffic counts of flow, speed, and headway used as inputs. Multiple simulation runs are then performed to obtain simulation outputs comparable to the real-world values. If the error is within the acceptable range, this indicates that the simulation model has been properly validated. Should the traffic model not be properly validated, the calibration and confirmation procedures are repeated until the differences between simulated and observed values are within an acceptable range. To quantify the model calibration performances, the Geoffrey E. Havers (GEH) statistic is used. GEH is defined as:(10)$$\mathrm{GEH}=\sqrt{\frac{2{\left(M-C\right)}^{2}}{M+C}}$$
where M is the simulated values and C is real-world values. Based on the guidelines by the Federal Highway Administration (FHWA) [32], a GEH lower than 5 for more than 85% of the observed pairs is acceptable for using traffic flow and travel times as inputs.

Thirdly, the microsimulation model is linked to PreScan (GUI) over a component object model server (Matlab), where the surrounding traffic is simulated at the same time as the conflict vehicles. Every change in the PreScan GUI is associated with a change in the Matlab/Simulink software. This Matlab/Simulink interface allows the user to design, add, implement, or validate the developed traffic conflict prediction algorithm within the ego-vehicle through a compilation sheet. The compilation sheet is composed of the infrastructure, vehicle tracking information, display ports, and the sensors in the scenario developed in PreScan.

With every run, the output files are stored in a database for post-processing. A 3D visualization viewer allows the user to analyze the results of the experiment by providing multiple viewpoints, pictures, and video generation capabilities. A summary of these stages is presented in Figure 5. The evaluation of the traffic conflict algorithm was carried out by analyzing the output of the developed algorithms. An AUC value for each simulation run was obtained by comparing the traffic conflict predictions from the developed algorithm to the traffic conflicts ground truth data. The traffic conflicts ground truth data was obtained by applying the criteria developed in Section 3.1 on the simulated data.

#### 3.2.4. Demonstration Experiment

This section investigates the simulation experiments that are adopted to highlight the usefulness and flexibility of the framework. The motorway scenery, the scenario adopted for testing, and all the different parameters are presented in this section. It is important to note that since this work is carried out in the UK, the speed limits are given in miles per hour (mph) and the simulation assumes left-hand driving.

##### Scenery Description and Parameters

Scenario testing was carried out on a section of the M1 motorway between Junction 19–21. This section was designed from the actual real-world motorway and represented as scenery. This provides a major advantage over artificial scenery, since the environmental parameters were taken from an on-site inspection. The environmental parameters include the position of roadside elements, lane widths, and shoulder widths. Figure 6 shows three pictures of a section of the M1 motorway environment that was utilized to test and validate the algorithm together with the slip roads and hard shoulders. Figure 6 includes the actual representation and how it is represented in PreScan and PTV VISSIM.

All simulation parameters required to build the scenery are presented in Table 2. Some parameters were extracted from Google Maps, such as total number of lanes, if a hard shoulder is presented, and lane markings. On the other hand, lane width was approximated by the average lane widths of motorways.

##### Scenario Description and Parameters

This section presents how the derived scenarios were transferred to a sub-microscopic traffic simulation environment. It also includes the process of the evaluation of the traffic conflict prediction algorithm. The scenario was defined by the parameters of the ego-vehicle, the opponent road user, and the control systems. In this study, the opponent user performs either a harsh deceleration or a lane change in front of the ego-vehicle to create a conflict.

Ego-Vehicle:

PreScan vehicle library provides detailed information on the ego-vehicle, ranging from vehicle body characteristics to vehicle components such as engine, suspension, steering, tires, and brakes. The library allows the user to create a virtual vehicle which replicates the behavior of a real vehicle. However, there are vehicles already available in the library that can be employed. This method was utilized for this study (Figure 7). A Ford Focus was implemented in order to replicate the instrumented vehicle used to collect real-time data as best as possible.

A table containing all the important ego-vehicle parameters is presented in Table 3.

Sensors;

In PreScan, a number of sensors can be implemented on the ego-vehicle in order to mimic the same information gathered from the ego-vehicle. It is also possible to modify the range and angle of view of the sensor in order to replicate similar data. PreScan distinguishes the sensors into three groups of physical models: idealized, detailed, and ground truth.

The idealized models replicate sensors that gather ground truth data of all the significant information needed from objects, such as position, size, type, and velocity. These sensors are used as a source from which the traffic-related factors can be estimated. They are positioned at approximately 400 m apart to mimic the Inductive Loop Detector (ILD) data within the motorway.

The detailed sensors provide the closest replication of data from real-world sensors. This is because these sensors allow the modification of multi-path effects or erroneous signals. Currently, PreScan provides high-fidelity sensors such as camera and radar, which are the sensors adopted in this study. Based on the sensor characteristics, these sensors provide an object list.

Ground truth sensors contribute by providing raw signals. For example, these sensors provide the output of a lidar instead of a list of objects. Moreover, all of the in-vehicle factors can be estimated using the self-demultiplexer in Simulink, which provides all required information about the ego-vehicle. An illustration of GUI parametrization of sensors used in this research and the conversion of the sensors into blocks in the Simulink interface is represented in Figure 8. By using the data collected by sensors, control algorithms are developed in the Simulink interface to estimate the factors required as inputs to the developed algorithm.

Simulink control blocks

The presented integrated simulation framework uses the Simulink interface as a layer over PreScan. This layer allows the user to read and modify the signals that flow between PreScan and the models. PreScan for Simulink consists of subsystems composed of blocks consisting of driving maneuvers, vehicle control, and vehicle model. All are connected similarly as with other Simulink blocks. The block modified further in this study was the vehicle control model. Simulink blocks can also be created and added to the experiment; examples of Simulink blocks developed to validate the prediction algorithm are presented here:1.To validate the algorithm, the ground truth must be generated to obtain the overall performance of the algorithm. The criteria adopted in Section 3.1 were developed in blocks. Using these blocks, ground truth traffic conflicts based on virtual data were extracted. An example of a module determining if the time headway is less than 3 s to issue a trigger warning is shown in Figure 9. To obtain an estimate of the current time headway, the distance was divided by velocity. If the value was between 0 and 3 s, a trigger was issued.2.It is essential to determine whether the vehicle detected from the sensors are (i) in the same lane as the ego-vehicle or (ii) changing lanes by measuring the angle between both vehicles. If either is true, the signal goes into another block (Figure 9) to determine whether a trigger should be issued. This is represented in Figure 10.3.The vehicle’s factors, i.e., velocity and yaw rate, were obtained directly from the sensor implemented within the ego-vehicle. The acceleration was estimated by dividing the change in velocity with the timestep of the simulation. Figure 11 represents the logic of how the vehicle factors were estimated.4.An example of how to trigger harsh braking within the simulation is presented in Figure 12. This figure shows how the two vehicles are connected in the compilation sheet for the ego-vehicle to react when the preceding vehicle performs a random harsh deceleration.

## 4. Data

The data used to develop the traffic conflict predictor was collected from two datasets. One dataset includes the real-world minute level traffic data from ILDs. These detectors collect data when there is a change in field, i.e., when an object (mainly a vehicle) passes over them. When a driver drives over a loop sensor, the loop field changes, which permits the detection device to detect the presence of an object. Such measurements include vehicle speed, occupancy, flow, and headway. These measurements are obtained from the loop detector with the lowest Euclidean distance between the ego-vehicle and the loop detector.

The other dataset is made up of traffic measurements collected by a Loughborough University instrumented car. This vehicle is equipped with multiple sensors, namely a Continental ARS308-2 long range radar, a PointGrey Grasshopper 3 (GS3-U3-41C6C-C) camera, a Ublox NEO-M8L GPS, a Mobileye device, a weather sensor, and an Arduino microcontroller. Each sensor has different strengths and weaknesses; therefore, by integrating the sensor data, an enhanced result can be obtained. This is because the advantages of using one sensor can compensate for the shortcomings of another sensor.

It is important to note that the ego-vehicle in the virtual environment was designed with the same sensors adopted in the real world to train the algorithm to collect the same factors for traffic conflict prediction. The data from each sensor and from the inductive loop detectors are unified and synchronized in a central data integration architecture. This results in a more trustworthy and safe system due to an increase in integrity and robustness of the system, while decreasing the errors related to the sensor systems as compared with a single sensor system [33,34]. This study is limited to considering only lane-change and rear-end conflicts because of the sensors adopted.

### Simulation Model Calibration and Validation

The baseline traffic microsimulation model was calibrated and validated for the time period between 11:00 and 12:00 a.m. based on traffic data from January 2016 and February 2017. The data over the total period was averaged to create a dataset that contains one observation per minute. Some of the data parameters that can be extracted from the data input in the simulation model include traffic flow per minute, route choice percentages, fleet composition characteristics, speed distribution, and time headway distribution.

It is important to note that the traffic flow values obtained from the ILDs match the simulation vehicle input points. These values are used to accurately represent the traffic flow fluctuations and route choices. In fact, to program the vehicle route decisions, the percentage of vehicles exiting the motorway was also derived. This was carried out by identifying the corresponding inductive loop detectors of the motorway and comparing the upstream flow with the down-stream one. The corresponding percentages are then assigned to the corresponding vehicle routes and the turning decisions of the vehicles in the simulation software.

The data also contains vehicle fleet composition information. Since the percentage of HGVs could not be varied temporally within the simulation, an overall average value 2qw calculated from the data. When considering the average speed and average time headway values, the same aggregation procedure was followed, but instead of per minute values, a statistical distribution was followed. These distributions were averaged over all detectors.

The measures of performance adopted for calibration are travel time and traffic flow values (following the guidelines provided by FHWA) [32]. For the real-world values, the travel time was estimated as the product of the average speed and the total distance traveled, while that of the simulated vehicles were collected from VISSIM. To calibrate traffic volume values, the GEH statistic was employed (presented in Equation 10) and all values were lower than 5. As a result, no alterations were made to the default driving behavior parameters of VISSIM.

## 5. Results

To prove the effectiveness of a traffic conflict predictor, testing and validation is essential. This was demonstrated in a virtual simulation environment. It is important to note that the algorithm was not trained on the virtual simulation data, but on the real-time data collected from the instrumented vehicle. PreScan (the sub-microscopic traffic simulator) provided an appropriate testbed to validate and test the traffic conflict predictions from the developed algorithm. PTV VISSIM was utilized to generate dynamic traffic surrounding the ego-vehicle. By implementing the prediction algorithm for the ego-vehicle in the Simulink interface, the algorithm was able to use the simulated data as an input and predict traffic conflicts or safe traffic dynamics surrounding the ego-vehicle in a virtual environment. However, the complexity of modeling all possible combinations of traffic situations and environmental conditions makes this approach challenging. Hence, a number of ‘benchmark’ scenarios had to be developed. These included (1) a rear-end conflict, (2) a lane change conflict, and (3) a combinational scenario. This section may be divided by subheadings. It should provide a concise and precise description of the experimental results and their interpretation, as well as the experimental conclusions that can be drawn.

### 5.1. Results of the Safety Performance Evaluation

This section presents the performances of the traffic conflict prediction algorithm. The virtual data produced was used to validate and test the predictions generated by the algorithm. Examples of particular simulation instances predicted as a (1) rear-end conflict, (2) lane change conflict, and (3) safe traffic dynamics scenario by the algorithm are presented. Each section describes the parameters of the ego-vehicle and the opponent vehicle during this instance. The parameters include the time headway, velocity, acceleration, and distance parameters. This was undertaken to determine whether these parameters satisfy the criteria established in Section 3.1. Finally, the overall results in terms of Area Under Curve (AUC) value, sensitivity, and accuracy of the developed algorithm are presented.

1.Example Evaluation of a Rear-End Conflict;

This section presents an example of a predicted rear-end conflict during one simulation run. It demonstrates an example of how the parameters were changed, during which a rear-end conflict was predicted. This scenario is presented in Figure 13.

In this example, the preceding vehicle performs a harsh deceleration in front of the ego-vehicle. During the predicted conflict, the time headway of the ego-vehicle is below 3 s at certain instances and the velocity and acceleration of both the ego-vehicle and the preceding vehicle were examined. In this instance, the ego-vehicle satisfied a criterion established in Section 3.1 for a rear-end conflict. For this example, v_EV_ > v_PV_ and the preceding vehicle exerts a large deceleration force when t_HEADWAY_ < 3 s. These parameters are depicted in Figure 14.

2.Example Evaluation of a Lane Change Conflict;

This section presents an example of a predicted lane change conflict. It demonstrates an example of how the parameters changed during a lane change conflict prediction. This scenario is presented in Figure 15.

In this example, the lane changing vehicle changes lane abruptly in front of the ego-vehicle. Similarly, during the predicted conflict, the time headway of the ego-vehicle is also below 3 s at certain instances. The velocity and acceleration of both the ego-vehicle and the preceding vehicle were examined. Following the criteria in Section 3.1, these criteria were also met. For this example, a_EV_ > v_PV_, a_EV_ ↑ when t_HEADWAY_ < 3 s. Some of the parameters were depicted in Figure 16. It is also important to note that the criterion of s_LONGITUDINAL_ < 75 m was met throughout this whole example.

3.Example Evaluation of a Safe Traffic Dynamic Scenario.

In this example, the simulation instance was chosen from a situation predicted as safe. This scenario is presented in Figure 17.

None of the thresholds for a conflict criterion were exceeded. In this experiment, although the opponent vehicle starts a smooth deceleration because of slight congestion ahead, the ego-vehicle has enough time headway to react to the smooth deceleration from the preceding vehicle. The algorithm did not predict this as a conflict. These parameters are depicted in Figure 18.

The algorithm predicted this instance as safe. In fact, neither vehicle exceeded the conflict criteria. The preceding vehicle smoothly decelerated because of congestion ahead and the ego-vehicle followed suit. However, there is no critical event.

### 5.2. Overall Results

The optimal traffic conflict prediction algorithm was implemented within the ego-vehicle in the Simulink interface. Approximately 25 conflicts were manually developed in PreScan for each simulation run. However, other conflicts also arose from the randomly-generated surrounding traffic dynamics. Each scenario was simulated 100 times by utilizing 100 different random seeds. The ground truth data was generated by applying the criteria in Simulink blocks. This was required in order to compare with the actual predictions from the developed algorithm. By utilizing the evaluation script, the output of the algorithm predictions was compared to the ground truth data and the overall performance metrics of the algorithm in each scenario--(1) the preceding vehicle performing harsh braking preceding the ego-vehicle, (2) a lane changing vehicle cuts in front of the ego-vehicle, and (3) a combination of both scenarios—are presented in Table 4.

Table 4 shows that the AUC values from each derived scenario in PreScan are relatively high, showing that the model has the ability to identify patterns that predict traffic conflicts. This demonstrates that the model is capable of using complex and non-linear patterns to classify and predict traffic conflicts based on a novel data set, from which the model was not trained. The threshold value was maintained ‘constant’ when the algorithm was developed (i.e., 0.389). The sensitivity values for this threshold are lower than when applied to the data on which the algorithm was trained, as expected. However, the algorithm had the ability to predict relatively sensitive results, especially in Scenario 1. For Scenario 1, for a FAR of 5–10%, the algorithm was able to predict 62.2–79.7% of traffic conflicts. When comparing Scenario 1 to Scenario 2, the algorithm was more sensitive in predicting rear-end conflicts. In Scenario 2, it was able to predict 73–79% of lane change conflicts for a 5–10% false alarm rate. In Scenario 3, it was also more sensitive than Scenario 2. This is attributed to fewer lane change conflicts on which the algorithm trained. Overall, even though the algorithm was applied to novel data, high sensitivity values were obtained.

## 6. Discussion

A plethora of studies have stated the importance of emerging vehicle technology for safety on roads. However, most of these claims are not quantitatively supported [35]. Virtual vehicle testing and validation is the key to addressing this gap in the research. Simulation framework has developed with different scenarios simulated under various combinations of parameters. Exploration of this area is not cost effective in real-world tests, as it imposes huge cost in the development cycle [17].

Several projects have assessed the safety performance of vehicle technologies in a virtual environment [36,37,38,39]. However, testing measures have not been standardized to date. It is challenging to assess different vehicle technologies, as numerous scenarios need to be developed. In this research, the traffic conflict prediction algorithm was validated based on the identification of relevant existing risky situations linked to rear-end and lane change conflicts. The scenario considered static attributes such as road design and dynamic content such as surrounding vehicles, their trajectories, and their behaviors. The results derived from the framework can be used as a pre-stage or parallel activity to field operational tests on public roads. To the author’s knowledge, no research has been published that uses an integrated framework to allow the assessment of vehicle-based traffic conflict algorithms on a motorway scenario.

Traffic scenarios were developed in the simulation framework to evaluate and validate the traffic conflict algorithm based on key conditions related to motorway operations. The safety performance evaluation showed that the algorithm obtained a high AUC value of 0.92, 0.88, and 0.90 for rear-end, lane-change, and a combination of both conflicts, respectively. The validation results of the algorithm were also significant due to high sensitivity and a low FAR, from which 80–84% and 73–79% of the rear-end and lane change traffic conflict were predictable for 10–20% FAR, respectively. This demonstrated that the model was capable of using complex and nonlinear patterns to classify and predict traffic conflicts based on a novel data se, on which the algorithm was not trained. The sensitivity values were also high, demonstrating the vehicle-based algorithm’s significant ability to predict traffic conflicts. Moreover, it is worth noting that the algorithm is more sensitive to rear-end conflicts, which could be due to the higher number of these events in the algorithm training.

There is an ongoing effort in the automotive industry to develop robust and reliable algorithms to estimate the threat assessment surrounding the ego-vehicle. These algorithms are key to ensuring safety and efficient operations for future intelligent mobility. The virtual experiments conducted within this research quantified the safety performance and prediction capabilities of the developed traffic conflict algorithm. Three scenarios were tested to represent (1) lane change conflicts, (2) rear-end conflicts, and (3) a combination of both. Each simulation run was defined by different random seeds, which generated different traffic dynamics surrounding the ego-vehicle. Parameters such as road design, driving maneuvers, and behavior of the ego-vehicle were kept constant. The variable parameters included the lateral and longitudinal position of the opponent vehicle and the velocity of the ego-vehicle. Changing parameters involve high computational requirements; hence only three essential parameters were varied. These simulation runs took approximately 4 min to be completed and the evaluation script took on average 30 s to be computed. In total, 100 simulations were conducted for each scenario, resulting in a total of 300 simulations to complete all the calculations. Though this process was time consuming, improving the computational time was not the focus. This set of parameters was sufficient to assess the performance of the vehicle-based traffic conflict algorithms.

To evaluate the prediction ability of traffic conflict algorithms in the integrated framework, the mean of the AUC values obtained from each simulation run was calculated. However, to acquire these AUC values, traffic conflicts ‘ground truth’ data from the simulation was required. This ‘ground truth’ data was obtained by the same criteria that were adopted in Section 3.1. It is important to note that the traffic conflicts in the simulation scenario were easier to extract compared to real-time data, since the vehicles’ trajectories and velocities were all known. In addition, the sensors’ detection of objects in the simulation has an 100% accuracy rate. This implies an ideal system, which is not the case in a real-world scenario because of errors associated with the sensor. Therefore, to accurately reproduce real-world behavior and the limitations of the sensors, some of the data applied to the developed prediction algorithm was carried out using high-fidelity sensors.

The integrated simulation framework developed in this research was based on a real-world motorway configuration in the UK. Although this is harder to implement, it has more benefits than adopting an artificial, virtual motorway. This is because static information for the scenario development such as lane widths, hard shoulders, and gradients can be taken from on-site observations. The dynamic information provided by the PreScan’s GUI such as brakes, tires, and suspension system is part of well-recognized reference models. These reference models were all validated and represented in blocks. Each block can be enhanced, transferred, or even replaced by other modules as long as they can provide the same output quantities. Other blocks can also be developed based on the aim of the simulation experiment. For example, the criteria used to extract the traffic conflict ground truth data was also developed in a self-contained block. These blocks have future implications for intelligent mobility, as they are transferable and can be readily applied by other studies.

While the traffic simulation model within VISSIM was intensively calibrated and validated using real-world data, some results may still be biased due to the underlying assumptions of this software. When considering the sub-microscopic simulation environment, one main limitation is that the scenarios developed relate to motorway operations and are based on human-related accident situations. For example, the scenarios developed might not reflect the new critical situations faced by intelligent vehicles. It is also important to mention that the safety risks due to mixed traffic, i.e., both intelligent and driver-operated vehicles, are still subject to research. For example, intelligent vehicles make use of onboard sensors to determine their environment, but still face a number of limitations in an urban environment, inclement weather conditions, or in a situation with unexpected behavior of traffic participants. These challenges are not explored in this study.

## 7. Conclusions

Testing and validating the developed traffic conflict prediction algorithms are the key to prove their effectiveness. However, the complexity of modeling all possible combinations of traffic situations and environmental conditions make this approach challenging. To validate the safety performance and prediction accuracy, the traffic conflict prediction algorithm was placed in the ego-vehicle in an integrated simulation framework. The framework consists of a sub-microscopic simulator, which provides an appropriate test bed to develop different scenarios to test and validate the algorithm, and a microscopic traffic simulation tool to simulate the surrounding traffic accurately based on real-time data. Rear-end and lane change traffic conflict scenarios were developed in the simulation framework to test and validate the effectiveness of the algorithm. The validation results from the integrated simulation framework are significant due to high sensitivity and a low FAR obtained for each scenario. As a result, this algorithm has the potential to be used in ADAS systems to develop proactive safety management strategies for improving traffic safety, presenting a viable solution for implementation within CAVs.

Though this framework provided several advantages for testing and validation, the developed scenarios need regular updating with new knowledge and data. This is because the generated scenarios might change over time, as road user behavior may alter with the increase in the market penetration rate of intelligent vehicles. It is unclear whether new critical situations may arise as a result, such as data inaccurately interpreted in a mixed traffic situation or in complex urban environments.

## Figures and Tables

**Figure 1 sensors-22-00566-f001:**
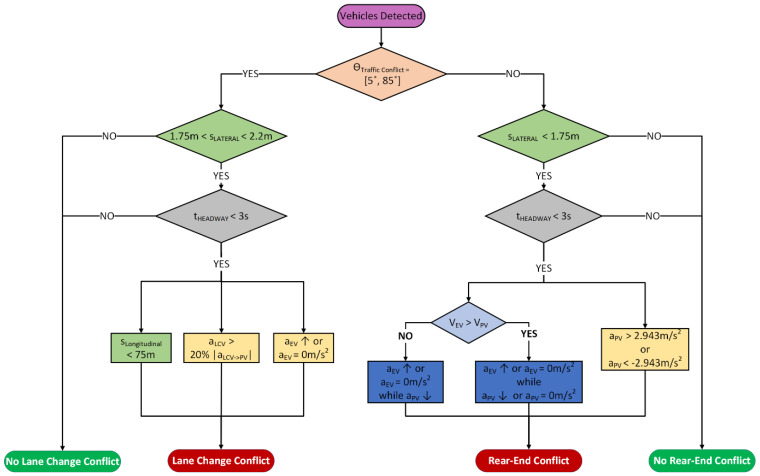
Automatic extraction algorithm for lane change and rear-end conflicts from video where EV, PV, and LCV represent the ego-vehicle, preceding vehicle, and lane change vehicle, respectively, while θ, t, s, V, and a represent angle, time, distance, velocity, and acceleration, respectively. Based on these criteria, the timestamps of each identified traffic conflict were recorded and validated. When the threshold was exceeded, the timestamp was highlighted as a traffic conflict by the system.

**Figure 2 sensors-22-00566-f002:**
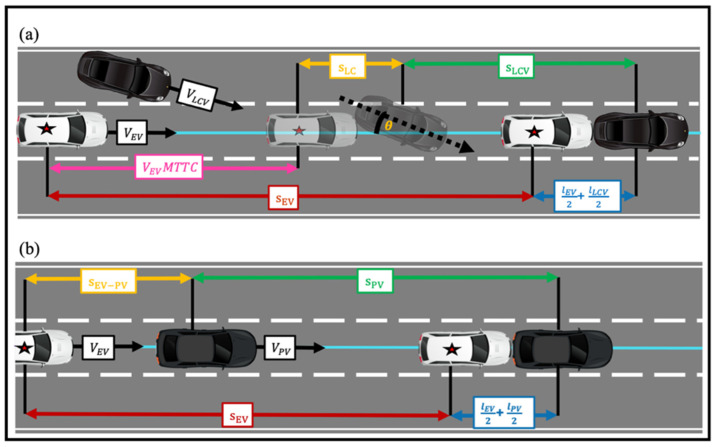
Visual representation of validating (**a**) lane change conflict and (**b**) rear-end conflict.

**Figure 3 sensors-22-00566-f003:**
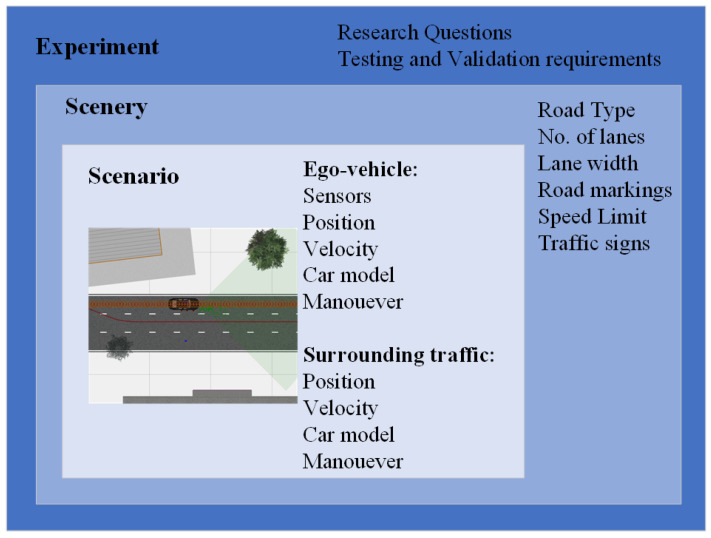
Ontology and hierarchy for simulation experiments adapted from [30].

**Figure 4 sensors-22-00566-f004:**
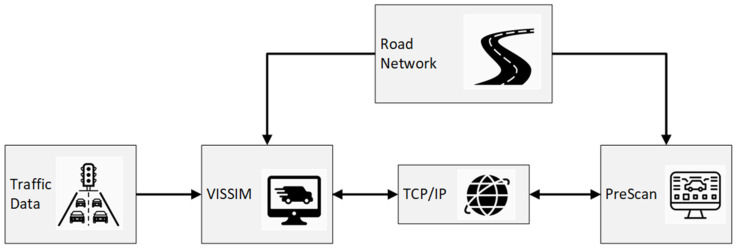
The architecture of the integrated simulation platform.

**Figure 5 sensors-22-00566-f005:**
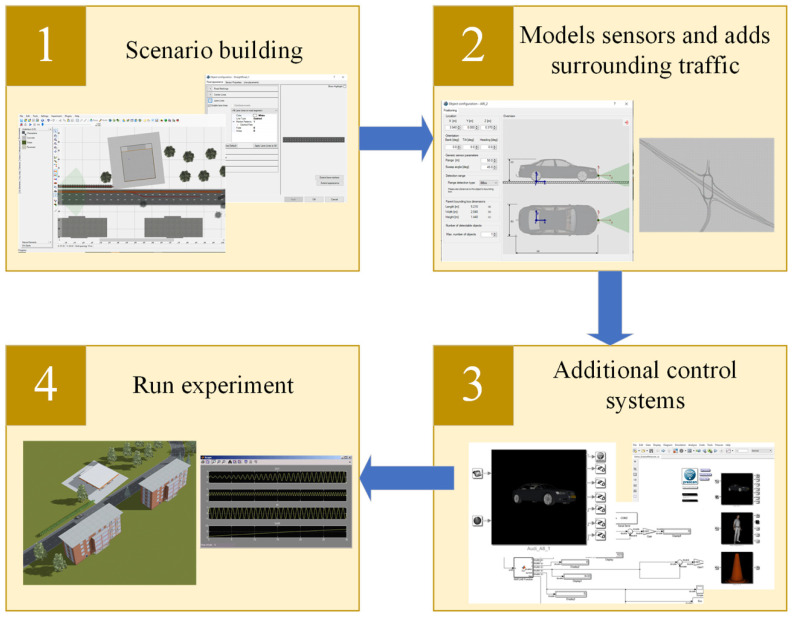
Developing stages of integrated simulation platform.

**Figure 6 sensors-22-00566-f006:**
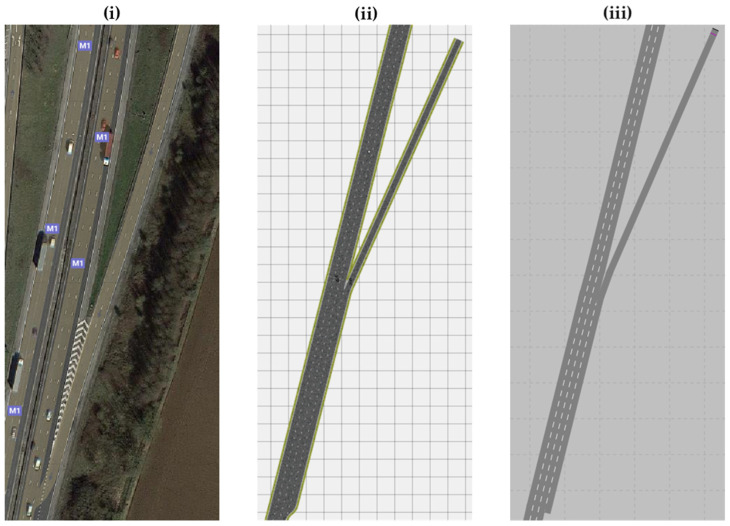
Section of the motorway for the demonstration experiment showing (**i**) actual representation and virtual reconstruction in (**ii**) PreScan and in (**iii**) PTV VISSIM.

**Figure 7 sensors-22-00566-f007:**
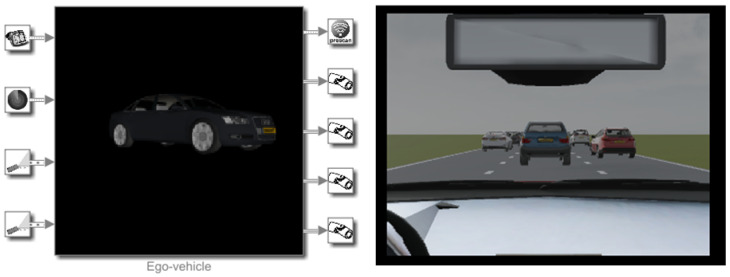
The ego-vehicle used in the simulation framework.

**Figure 8 sensors-22-00566-f008:**
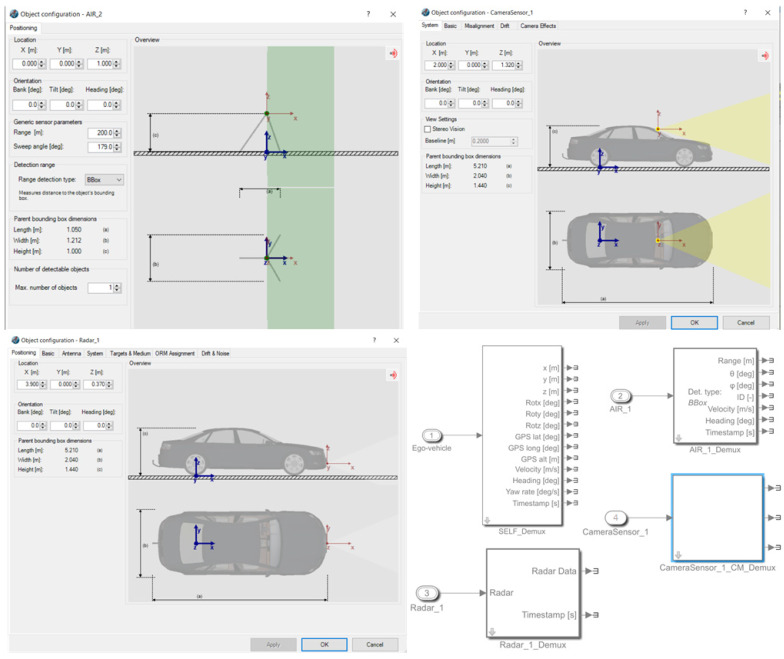
Sensor’s parameters in PreScan GUI converted to module blocks in Simulink interface (bottom right).

**Figure 9 sensors-22-00566-f009:**
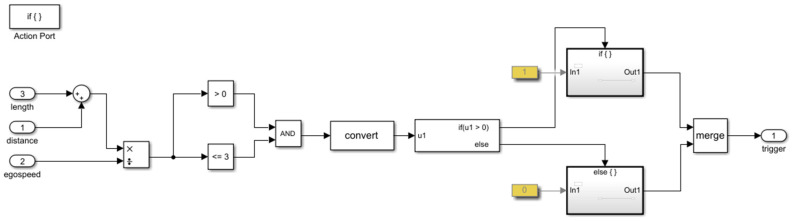
Time headway trigger warning in Simulink.

**Figure 10 sensors-22-00566-f010:**
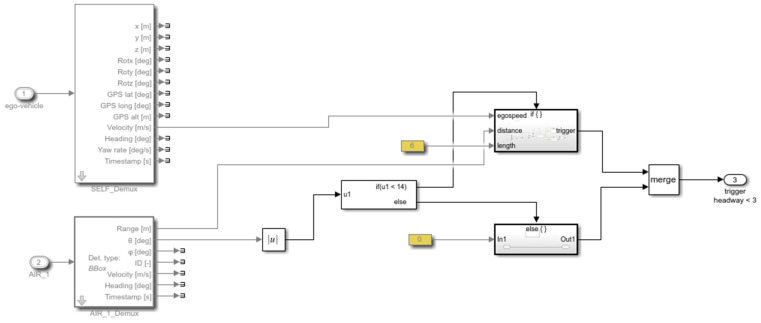
Time headway trigger for preceding vehicle in Simulink.

**Figure 11 sensors-22-00566-f011:**
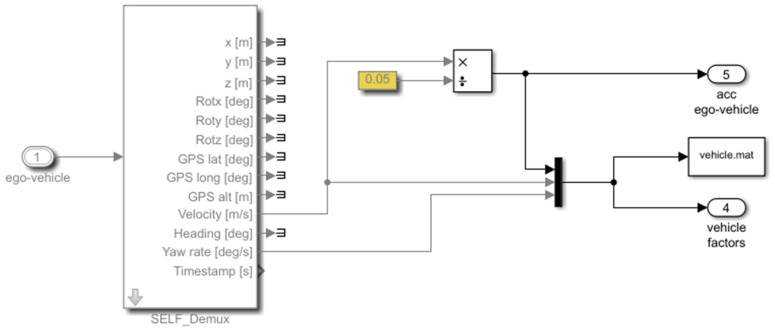
Estimation of vehicle-related factors in Simulink.

**Figure 12 sensors-22-00566-f012:**
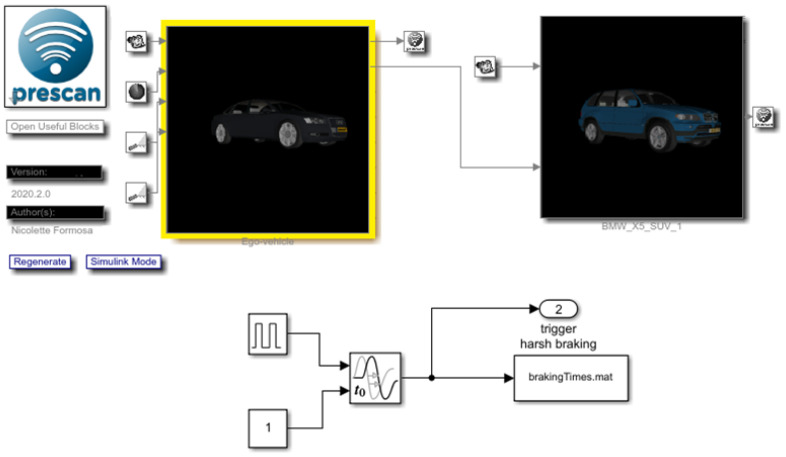
Connection of ego-vehicle to preceding vehicle and harsh braking triggering.

**Figure 13 sensors-22-00566-f013:**
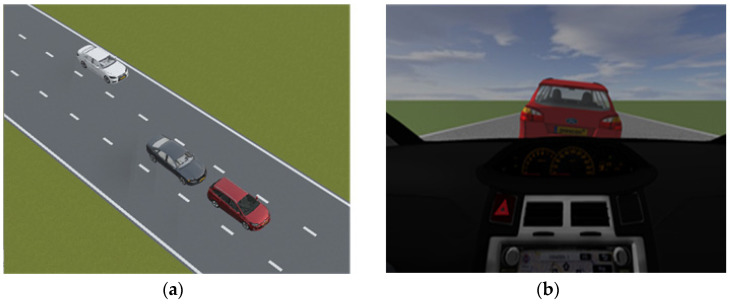
Rear-end conflict in the simulation environment from (**a**) top-view and (**b**) from driver’s view.

**Figure 14 sensors-22-00566-f014:**
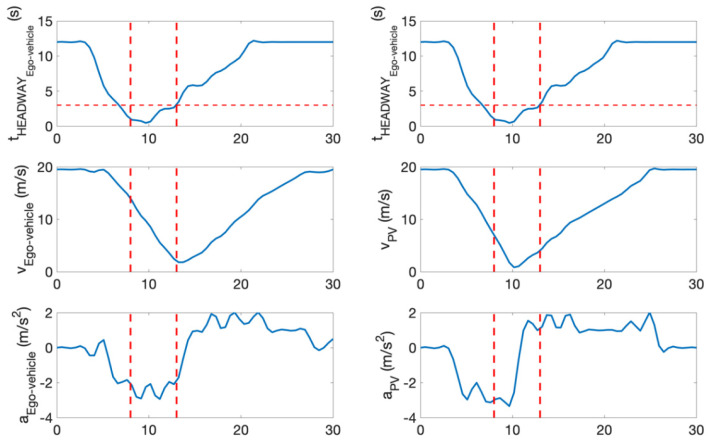
Time headway of ego-vehicle, velocity, and acceleration of ego-vehicle and opponent vehicle.

**Figure 15 sensors-22-00566-f015:**
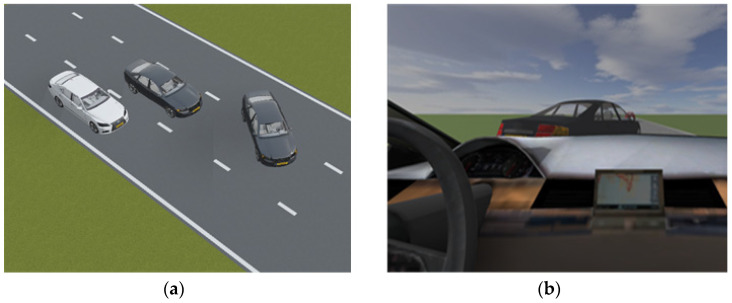
Lane change conflict in the simulation environment from (**a**) top-view and from (**b**) driver’s view.

**Figure 16 sensors-22-00566-f016:**
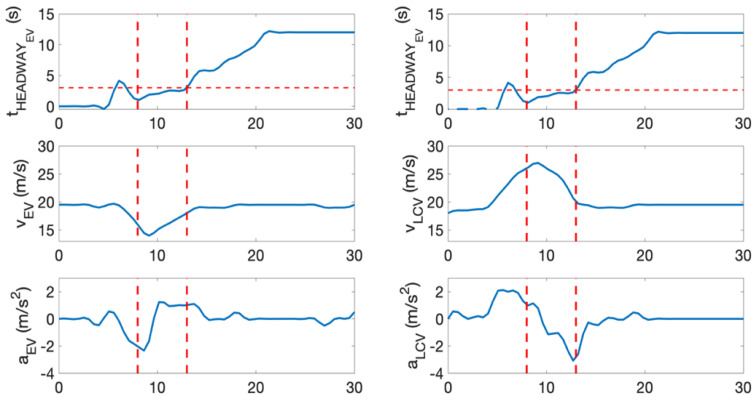
Time headway of ego-vehicle, velocity, and acceleration of ego-vehicle and opponent vehicle.

**Figure 17 sensors-22-00566-f017:**
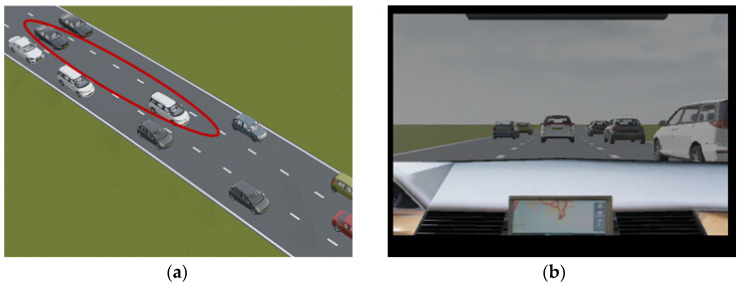
Safe Traffic Dynamic Scenario in the simulation environment from (**a**) top-view and from (**b**) driver’s view.

**Figure 18 sensors-22-00566-f018:**
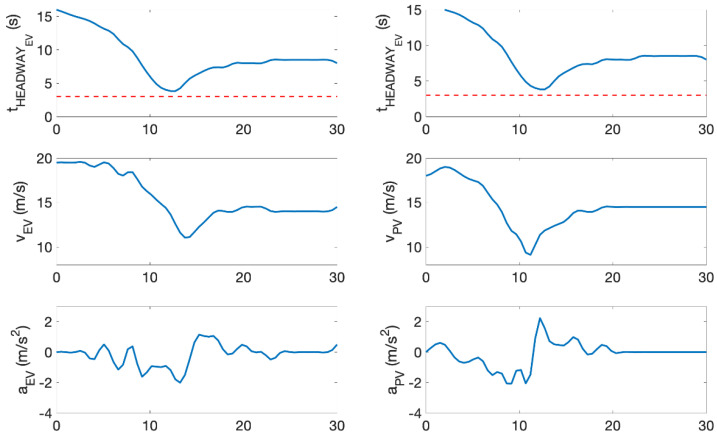
Time headway of ego-vehicle, velocity, and acceleration of ego-vehicle and opponent vehicle.

**Table 1 sensors-22-00566-t001:** Criteria for the selection of the appropriate sub-microscopic simulation tool (adapted from [17]).

Requirement	Satisfied (Yes/Partially/No)		
	PreScan (TASS International)	CarMaker (IPG)	CARLA (Intel)
Open Source	No	No	Yes
Environmental sensor models	Yes	Partially	Partially
Road and roadside models	Yes	Yes	Partially
Realistic vehicle models to generate different vehicles	Partially	Yes	Partially
Pedestrian and bicycle models	Yes	Partially	Yes
Realistic driver or maneuver modeling	Partially	Yes	No
Adverse weather conditions can be adopted and also affect the sensor models	Yes	No	No
Allows scripting and automation functionality which links to other platforms	Yes	Yes	Yes
Potential to change visibility within the experiment, e.g., darkness and retro-reflectivity	Yes	Partially	No
Potential to simulate worn off traffic signs and faded lane markings, (perceived by sensor models)	Yes	Partially	No
Allows road infrastructure data to be imported and (semi-) automatic generation of road models	Partially	Partially	Partially
Needs additional software tools (e.g., Matlab/Simulink/Python)	Yes	Yes	Yes

**Table 2 sensors-22-00566-t002:** Simulation parameter setup related to the scenery.

Road Scenery Parameters	Values
Area	Motorway
Number of lanes	3–4
Road markings	Yes
Lane width	3.65 m
Roadside	Shoulder near slip roads
Speed limit	70 mph
Light conditions	Daylight
Horizontal geometry	Straight

**Table 3 sensors-22-00566-t003:** Ego-vehicle’s parameter setup.

Vehicle Model Parameters	Value
Vehicle model	Ford Focus
Transmission	Automatic
Mass	1471 kg
Reference area	2.74 m^2^
Max acceleration	2.934 m/s^2^
Max deceleration	−2.934 m/s^2^
Roll coefficient	0.01
Drag coefficient	0.31

**Table 4 sensors-22-00566-t004:** Validation from PreScan of traffic conflict prediction algorithm in a virtual environment.

**(1) Scenario 1—Preceding Vehicle Performs Harsh Deceleration**			
**Average Sensitivity**	**FAR**	**Average Accuracy**	**Average AUC Value**
0.622	5.0%	0.903	0.916
0.797	10.0%	0.844	
0.843	20.0%	0.783	
0.925	30.0%	0.682	
**(2) Scenario 2—Lane Changing Vehicle Cuts in before the Ego-Vehicle**			
**Average Sensitivity**	**FAR**	**Average Accuracy**	**Average AUC Value**
0.573	5.0%	0.853	0.883
0.730	10.0%	0.819	
0.785	20.0%	0.764	
0.834	30.0%	0.673	
**(3) Scenario 3—Combination of Both Scenarios**			
**Average Sensitivity**	**FAR**	**Average Accuracy**	**Average AUC Value**
0.607	5.0%	0.875	0.901
0.774	10.0%	0.839	
0.812	20.0%	0.782	
0.867	30.0%	0.653	

## Data Availability

The data presented in this study are available on request from the corresponding author.

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
