# Peer review of "Validating a Traffic Conflict Prediction Technique for Motorways Using a Simulation Approachâ€"

_sensors, 2022, doi:10.3390/s22020566_

Round 1
Reviewer 1 Report
Dear Sirs,
the article presents an important aspect of collision mitigation for the autonomous vehicles, which is a very popular problem right now.
I do not have any significant comments, maybe besides such minorities as:
- line 396 - maybe GEH should be explained in brackets();
- maybe figs 9, 10 and 11 should be more clearly explained
Author Response
Sensors-1511366
Validating a Traffic Conflict Prediction Technique for Motorways using a Simulation approach
Response to the referees’ comments
Formosa, N., Quddus, M., Alkis Papadoulis, Andrew Timmis
Thank you for providing us with the opportunity to further revise our paper titled ‘Validating a Traffic Conflict Prediction Technique for Motorways using a Simulation approach’ to Sensors. We have now addressed a few ‘minor’ comments and incorporated them in our revised paper. Please see below, in blue, our detailed response to each of the comments and in red what was added to the revised manuscript.
Reviewer comment [1.1]: The article presents an important aspect of collision mitigation for the autonomous vehicles, which is a very popular problem right now.
I do not have any significant comments, maybe besides such minorities as:
- line 396 - maybe GEH should be explained in brackets();
- maybe figs 9, 10 and 11 should be more clearly explained
Authors’ response [1.1]: Thank you very much for your positive evaluation of our paper. We also appreciate the referee’s efforts. We have addressed the minorities mentioned by reviewer as:
- Line 396 now line 739 was modified to ‘To quantify the model calibration performances, Geoffrey E. Havers (GEH) statistic is used’.
- A clearer explanation of the figures mentioned by the reviewer are given as:
- Figure 9 (line 849-855)
‘To validate the algorithm, the ground truth must be generated to obtain the overall performance of the algorithm. The criteria adopted in Section 3.1 were developed in blocks. Using these blocks, ground truth traffic conflicts based on virtual data were extracted. An example of a module determining if the time headway is less than 3 seconds to issue a trigger warning, is shown in Figure 9. To obtain an estimate of the current time headway, the distance was divided by velocity. If the value was between 0 and 3 seconds, a trigger was issued.’
- Figure 10 (line 858-862)
‘It is essential to determine whether the vehicle detected from the sensors are (i) in the same lane as the ego-vehicle or (ii) changing lanes, by measuring the angle between both vehicles. If either one was true, the signal goes into another block (Figure 9) to determine whether a trigger should be issued. This is represented in Figure 10.’
- Figure 11 (line 865-868)
‘The vehicle's factors, i.e., velocity and yaw rate were obtained directly from the sensor implemented within the ego-vehicle. The acceleration was estimated by dividing the change in velocity with the timestep of the simulation. Figure 11 represents the logic how the vehicle factors were estimated.’
- Figure 12 (line 880-883)
‘An example of how to trigger harsh braking within the simulation is presented in Figure 12. This figure shows how the two vehicles are connected in the compilation sheet for the ego-vehicle to react when the preceding vehicle performs a random harsh deceleration.’
Reviewer 2 Report
Keywords: A narrow set of keywords that won’t necessarily lead to “hits” when other researchers search for research from your field.
Introduction:
The authors should better highlight the problems or Research Gap. What new will this research bring?
Literature Review:
Line 97: “It must be noted that only lane change and rear-end traffic conflicts are considered in this study, as these are the two most prevalent type of traffic conflicts in this setting”. This claim is related to the source: Najafabadi MM, Villanustre F, Khoshgoftaar TM, Seliya N, Wald R, Muharemagic E. Deep learning applications and challenges in big data analytics. J Big Data. 2015;2(1):1–21.. Is this a “justifiable” limitation of this research?
Line 131: “This approach is typically followed in existing studies for intersections using static video data.” What research. What are the findings?
The literature review does not include a review of comparable research and findings. And then at the end there is a need to represent Research Gap (hypothesis) that would justify the need for this research.
The article is “popularly” written (and not scientifically) in certain parts. The article is generally well written, in some places too long.
Author Response
Sensors-1511366
Validating a Traffic Conflict Prediction Technique for Motorways using a Simulation approach
Response to the referees’ comments
Formosa, N., Quddus, M., Alkis Papadoulis, Andrew Timmis
Thank you for providing us with the opportunity to further revise our paper titled ‘Validating a Traffic Conflict Prediction Technique for Motorways using a Simulation approach’ to Sensors. We have now addressed a few ‘minor’ comments and incorporated them in our revised paper. Please see below, in blue, our detailed response to each of the comments and in red what was added to the revised manuscript.
Referee 2:
Reviewer comment [2.1]: Introduction: The authors should better highlight the problems or Research Gap. What new will this research bring?
Authors’ response [2.1]: Thank you for this comment.
The following was added to the introduction section:
Problems:
- line 56-60: ‘Typically, a traffic conflict includes including at least two moving vehicles and is generally identified through the response of the evasive action that takes place so as to avoid a potential collision and by a safety surrogate measure (SSM) which shows a spatio-temporal risk (two parties are in close proximity in either space or time) [5–7]. Nevertheless, reliable traffic conflict identification is not an easy task.’
- Line 65-67: ‘To determine the prediction accuracy and impact on safety the traffic conflict prediction algorithm can achieve in a complex and organically changing real-world environment, testing and validation is the key for facilitating customer uptake.’
Research Gap (line 80-103):
‘Therefore, the purpose of this paper is to validate a traffic conflict prediction algorithm implemented within an integrated sub-microscopic simulation framework. Addressing one main challenge arising from validating the prediction algorithm is that ground truth, i.e., traffic conflicts, is not readily available. This study makes use of a system which adopts a set of defined conservative criteria derived from existing literature to identify all the possible traffic conflicts. To enhance the validation of ground truth, the results from this system are also validated. The ground truth provides a means for virtual verification and evaluation of the traffic conflict predictions from the algorithm in critical scenarios using an integrated simulation framework. To ensure realistic values and accurate representation of traffic characteristics, data was collected via an instrumented vehicle and inductive loop detectors from a section of the M1 motorway in the UK. The scenarios are developed based on several variables including the ego-vehicle and the behavior of the surrounding vehicles, the sensors used and the driving style of road users. These variables need to be modelled in a sub-microscopic simulation environment while a microscopic traffic simulator can be used in conjunction to generate traffic dynamics.
This research is a valuable addition to existing literature, since it develops a process to acquire traffic conflict data to test the effectiveness of the prediction algorithm by developing a number of traffic conflict scenarios in the simulation framework. The integrated simulation framework is a new approach which provides a platform to the user to model, test, validate and further enhance the customized models for future users. The results emerging from the framework showcase transferability of the algorithm and provide recommendations to vehicle manufacturers to improve their collision-risk model allowing intelligent vehicles to operate safely and reliably and to further test and develop vehicle safety technologies and automated driving systems.’
Reviewer comment [2.2]: Line 97: “It must be noted that only lane change and rear-end traffic conflicts are considered in this study, as these are the two most prevalent type of traffic conflicts in this setting”. This claim is related to the source: Najafabadi MM, Villanustre F, Khoshgoftaar TM, Seliya N, Wald R, Muharemagic E. Deep learning applications and challenges in big data analytics. J Big Data. 2015;2(1):1–21. Is this a “justifiable” limitation of this research?
Authors’ response [2.2]: We think that this is justifiable because lane change and rear-end conflicts are most common type of conflicts on a motorway setting as stated by the reference mentioned by the reviewer. Moreover, the data used to carry out this research is also collected from a motorway setting. During the real-time data collection, these two types of conflicts were the only conflicts observed so these were adopted in this study.
Reviewer comment [2.3]: Line 131: “This approach is typically followed in existing studies for intersections using static video data.” What research. What are the findings?
Authors’ response [2.3]: This part of the literature review shows how traffic conflicts are identified in the literature. Two references were added in the manuscript to the sentence the reviewer mentioned as shown [15,16]. The findings in these existing studies use static video data cannot be adopted for an in-vehicle perspective research, therefore, cannot be adopted in connected and autonomous vehicles (CAVs). The following was added to the revised manuscript (line 305-312):
‘This approach is typically followed in existing studies for intersections using static video data [15, 16]. However, by using static video data, inherent challenges arising from the development of the latest technology advancements such as autonomous vehicles, are not covered. Video data collected from an in-vehicle perspective could add a new real-time dimension to traffic conflict identification analysis by examining a continuously changing environment. Therefore, such a dynamic video analysis system needs to be developed to provide valid and reliable traffic conflict identification for a continuously changing visual and driving environment.’
- Tageldin A, Sayed T, Ismail K. Evaluating the safety and operational impacts of left-turn bay extension at signalized intersections using automated video analysis. Accid Anal Prev. 2018;120:13–27
- Guo Y, Sayed T, Zaki MH, Transportation Research B. Automated Safety Diagnosis of Powered Two Wheelers at a Shared Traffic Facility in China. Transp Res Board, 96th Annu Meet. 2017;10:17
Reviewer comment [2.4]: The literature review does not include a review of comparable research and findings. And then at the end there is a need to represent Research Gap (hypothesis) that would justify the need for this research.
Authors’ response [2.4]: This comment was addressed as follows:
- The research in this context is very limited. Comparable research was reviewed with regards to other studies (lines 397-459) using sub-microscopic simulation to test different ADAS and a thorough review of each simulator is provided in Table 1.
- Research Gap line 490-506: ‘In conclusion, testing and validation of any ADAS is crucial. Existing studies have investigated the safety effectiveness of different ADAS functions in various simulators [19-25]. However, ground truth is not always known a-priori. In this study, to extract a potential traffic conflict, criteria derived from existing literature are used to include all traffic conflicts. Therefore, using these criteria an extraction algorithm is adopted to identify all time points where the thresholds were exceeded. Nevertheless, the time points identified by the system need to be validated. By comparing the extracted traffic conflicts ground truth with traffic conflicts predictions from the implemented algorithm, the technical reliability and performance of the algorithm can be evaluated. Such processes can be performed in an integrated simulation framework. In this research, the framework considered consists of a submicroscopic simulator (PreScan) which provides the appropriate testbed and can accurately simulate the components of an intelligent vehicle, in conjugation with a microscopic simulator (PTV VISSIM) to simulate the surrounding traffic dynamics. This framework is considered due to its wide acceptability within the research community and its capability to test ADAS. This work is significant as it provides benchmarking for the prediction of traffic conflicts for virtual testing procedures and is a relevant step for developing safety management strategies for AVs.’
Reviewer comment [2.5]: The article is “popularly” written (and not scientifically) in certain parts. The article is generally well written, in some places too long.
Authors’ response [2.5]: Thank you for this comment. The revised manuscript was reviewed, and the parts which the authors thought were long and not scientifically written were modified.
Reviewer 3 Report
Dear Authors,
thanks a lot for the manuscript, which is interesting and sound. Despite this misalignment, I would like to give you the following feedback:
- I suggest explaining "microscopic" and "submicroscopic" in the abstract for the readers with a non-traffic simulation background
- Please use [ ] instead of ( ) for the references.
- The sentence starting at line 46 seems to be incomplete
- line 57: => have been developed
- line 101: incomplete sentence?
- line 158: => equations to
- line 181-183: incomplete sentence?
- line 186: mixture of present and passt tense
- line 195: emplys => employs (?)
- line 209: You should provide an explanation, why you have chosen PreScan for you study
- line 217ff: please provide references for the statement that "Existing studies have investigated..."
- line 244-247: sentence incomplete?
- line 254: delete one "conflicts"
- caption of fig. 1 should be on the same page as the figure.
- equations (1), (2), (3), (5) and (8) should provide more line breaks as they exceed into the margin.
- line 286: please provide a space between the value and its unit (=> 0 m)
- In line 355 you state that the two simulators exchange data via TCP/IP, but the arrows in fig. 4 are oneway arrows.
- You rely on the reader's experience with Matlab/Simulink in fig 9 ff. I suggest explaining it little bit more for this readers who are not familiar with Simulink.
- line 524 permit => permits (?)
Author Response
Sensors-1511366
Validating a Traffic Conflict Prediction Technique for Motorways using a Simulation approach
Response to the referees’ comments
Formosa, N., Quddus, M., Alkis Papadoulis, Andrew Timmis
Thank you for providing us with the opportunity to further revise our paper titled ‘Validating a Traffic Conflict Prediction Technique for Motorways using a Simulation approach’ to Sensors. We have now addressed a few ‘minor’ comments and incorporated them in our revised paper. Please see below, in blue, our detailed response to each of the comments and in red what was added to the revised manuscript.
Reviewer comment [3.1]: Thanks a lot for the manuscript, which is interesting and sound. Despite this misalignment, I would like to give you the following feedback:
Authors’ response [3.1]: Thank you very much for your positive feedback.
Reviewer comment [3.2]: I suggest explaining "microscopic" and "submicroscopic" in the abstract for the readers with a non-traffic simulation background
Authors’ response [3.2]: Thank you for this comment. The following sentence was modified in the abstract to reflect the definition of "microscopic" and "submicroscopic" in the abstract.
‘This framework consisted of a submicroscopic simulator, which provided an appropriate testbed to accurately simulate the components of an intelligent vehicle and a microscopic traffic simulator able to generate the surrounding traffic.’
Reviewer comment [3.3]: Please use [ ] instead of ( ) for the references.
Authors’ response [3.3]: Thank you, these brackets were changed accordingly.
Reviewer comment [3.4]: The sentence starting at line 46 seems to be incomplete
Authors’ response [3.4]: Thank you this sentence has now moved to line 47 and has been modified to:
‘The underlying theory behind the approach of using non-accident data is based on that in accelerating vehicle-based proactive safety systems, traffic conflicts can be used a measure of accident nearness and a safety critical event has the potential to progress to a traffic collision.’
Reviewer comment [3.5]: line 57: => have been developed
Authors’ response [3.5]: Thank you the verb has been modified to ‘have been developed’ (now line 62).
Reviewer comment [3.6]: line 101: incomplete sentence?
Authors’ response [3.6]: This sentence now in lines 274-276 was modified to:
‘Secondly, it presents a review of existing integrated simulation frameworks to test and validate a traffic conflict predictor algorithm. These frameworks assist to improve safety in ADASs and highlight their utility in CAVs and AVs.’
Reviewer comment [3.7]: line 158: => equations to
Authors’ response [3.7]: Thanks, a space was added (now line 377).
Reviewer comment [3.8]: line 181-183: incomplete sentence?
Authors’ response [3.8]: Thank you this sentence (now line 400-402) was modified to:
‘On the other hand, Seo et al., [20] adopted a virtual sensor model together with control algorithms, to compare autonomous emergency braking with a pre-safe seat belt system.’
Reviewer comment [3.9]: line 186: mixture of present and past tense
Authors’ response [3.9]: Thanks for pointing this out. This was modified in line 405-406 to:
‘Riegger et al. [21] developed and validated a system for safe crossing. The system took control over AVs to optimise their trajectories.’
Reviewer comment [3.10]: line 195: emplys => employs (?)
Authors’ response [3.10]: Thank you, this was changed to ‘employs’. (line 457).
Reviewer comment [3.11]: line 209: You should provide an explanation, why you have chosen PreScan for you study
Authors’ response [3.11]: Thanks for this comment. An explanation was added in line 470-489 on the revised manuscript and presented here:
‘As a result, from Table 1, PreScan was the simulation tool adopted for this study. This is because of its ability to integrate the essential multiple components of a traffic model, including vehicles (i.e., intelligent mobility functionalities), human behavior and traffic network via user interfaces. PreScan can also simulate multiple sensor technologies such as, cameras, LiDARs, radars and GPS. It operates by employing Simulink, enabling the representation of sophisticated system/scenarios control algorithms, development of new algorithms and modifications of existing ones. The main strength of this platform is its ability to offer a variety of intelligent traffic modelling in order to design and investigate various technologies and applications.’
Reviewer comment [3.12]: line 217: please provide references for the statement that "Existing studies have investigated..."
Authors’ response [3.12]: The following references were added to this statement.
‘Existing studies have investigated the safety effectiveness of different ADAS functions in various simulators [19-25].’
- Schubert R, Mattern N, Bours R. Simulation of Sensor Models for the Evaluation of Advanced Driver Assistance Systems. ATZelektronik Worldw. 2014;9(3):26–9.
- Seo B, Park Y, Han S, Battaglia S. Crashworthiness Improvement through Integrated Analysis of Active and Passive Safety Systems. Proc FISITA World Congr 2014 Maastricht, Netherlands. 2014;
- Riegger L, Carlander M, Lidander N, Murgovski N, Sjöberg J. Centralized MPC for autonomous intersection crossing. IEEE Conf Intell Transp Syst Proceedings, ITSC. 2016;1372–7.
- Überbacher M, Wolze P, Burtsche T. Experiencing Safety Function Testing. ATZ Worldw. 2017;119(7–8):54–7.
- Pfeffer R, Haselhoff M. Video Injection Methods in a Real-World Vehicle for Increasing Test Efficiency. ATZ Worldw. 2016;44–9.
- Osiński B, Miłoś P, Jakubowski A, Zięcina P, Martyniak M, Galias C, et al. CARLA Real Traffic Scenarios -- novel training ground and benchmark for autonomous driving. 2020;
- Dosovitskiy A, Ros G, Codevilla F, Lopez A, Koltun V. CARLA: An Open Urban Driving Simulator. 2017;(CoRL):1–16.
Reviewer comment [3.13]: line 244-247: sentence incomplete?
Authors’ response [3.13]: This sentence was modified in line 522-526 as:
“The submicroscopic simulator served as an appropriate testbed to validate and test the traffic conflict predictions from the developed algorithm implemented within the ego-vehicle. The microsimulation tool was utilized to simulate dynamic traffic surrounding the ego-vehicle.”
Reviewer comment [3.14]: line 254: delete one "conflicts"
Authors’ response [3.14]: Thank you for pointing this out. One “conflicts” was removed.
Reviewer comment [3.15]: Caption of fig. 1 should be on the same page as the figure.
Authors’ response [3.15]: Thanks, caption is now in the same page as the figure.
Reviewer comment [3.16]: Equations (1), (2), (3), (5) and (8) should provide more line breaks as they exceed into the margin.
Authors’ response [3.16]: These equations were modified in the revised manuscript to fit within the margins.
Reviewer comment [3.17]: line 286: please provide a space between the value and its unit (=> 0 m)
Authors’ response [3.17]: Thank you, a space was added between value and its unit. (line 618)
Reviewer comment [3.18]: In line 355 you state that the two simulators exchange data via TCP/IP, but the arrows in fig. 4 are one-way arrows.
Authors’ response [3.18]: Thank you, the reviewer is correct. Figure 4 was modified accordingly.
Reviewer comment [3.19]: You rely on the reader's experience with Matlab/Simulink in fig 9 ff. I suggest explaining it little bit more for this readers who are not familiar with Simulink.
Authors’ response [3.19]: We agree with the reviewer. An explanation for these was added in the revised manuscript and presented below:
- A clearer explanation of the figures mentioned by the reviewer are given as:
- Figure 9 (line 849-855)
‘To validate the algorithm, the ground truth must be generated to obtain the overall performance of the algorithm. The criteria adopted in Section 3.1 were developed in blocks. Using these blocks, ground truth traffic conflicts based on virtual data were extracted. An example of a module determining if the time headway is less than 3 seconds to issue a trigger warning, is shown in Figure 9. To obtain an estimate of the current time headway, the distance was divided by velocity. If the value was between 0 and 3 seconds, a trigger was issued.’
- Figure 10 (line 858-862)
‘It is essential to determine whether the vehicle detected from the sensors are (i) in the same lane as the ego-vehicle or (ii) changing lanes, by measuring the angle between both vehicles. If either one was true, the signal goes into another block (Figure 9) to determine whether a trigger should be issued. This is represented in Figure 10.’
- Figure 11 (line 865-868)
‘The vehicle's factors, i.e., velocity and yaw rate were obtained directly from the sensor implemented within the ego-vehicle. The acceleration was estimated by dividing the change in velocity with the timestep of the simulation. Figure 11 represents the logic how the vehicle factors were estimated.’
- Figure 12 (line 880-883)
‘An example of how to trigger harsh braking within the simulation is presented in Figure 12. This figure shows how the two vehicles are connected in the compilation sheet for the ego-vehicle to react when the preceding vehicle performs a random harsh deceleration.’
Reviewer comment [3.20]: line 524 permit => permits (?)
Authors’ response [3.20]: Thank you, this was changed to ‘permits’. (line 891)